# On the Universal Approximation Properties of Deep Neural Networks using MAM Neurons

## Abstract

As Deep Neural Networks (DNNs) are trained to perform tasks of increasing complexity, their size grows, presenting several challenges when it comes to deploying them on edge devices that have limited resources. To cope with this, a recently proposed approach hinges on substituting the classical Multiply-and-Accumulate (MAC) neurons in the hidden layers of a DNN with other neurons called Multiply-And-Max/min (MAM) whose selective behaviour helps identifying important interconnections and allows extremely aggressive pruning. Hybrid structures with MAC and MAM neurons promise a reduction in the number of interconnections that outperforms what can be achieved with MAC-only structures by more than an order of magnitude. However, by now, the lack of any theoretical demonstration of their ability to work as universal approximators limits their diffusion. Here, we take a first step in the theoretical characterization of the capabilities of MAM&MAC networks. In details, we prove two theorems that confirm that they are universal approximators providing that two hidden MAM layers are followed either by a MAC neuron without nonlinearity or by a normalized variant of the same. Approximation quality is measured either in terms of the first-order $L^p$ Sobolev norm or by the $L^\infty$ norm.

## 1 Introduction

Deep Neural Networks (DNNs) solve complex tasks leveraging a massive number of trainable parameters. Yet, thanks to the recent increasing interest in mobile Artificial Intelligence, there has been a growing emphasis on designing lightweight structures able to run on devices with constrained resources. This can be obtained by removing parameters that do not appreciably influence performance by means of one of the many pruning techniques that have been proposed. Some approaches entail removing, in a single shot, individual interconnections or entire neurons once the DNN has been trained, while others methods are applied iteratively, and require multiple rounds of training. These techniques eliminate interconnections but do not alter the underlying Multiply-and-ACcumulate (MAC) paradigm that governs the neuron's inner functioning.

In [1, 2], the authors address the challenge of designing neural networks that can have a smaller memory footprint presenting a novel neuron model based on the Multiply-And-Max/min (MAM) paradigm that can be substituted to classical MAC neurons in the hidden layers of a DNN to allow a more aggressive pruning of interconnections, while substantially preserving the network performance. In a standard MAC-based neuron, inputs are modulated independently of each other through multiplication with their respective weights, and the resulting products are then summed into a single quantity. As MAC neurons, MAM neurons multiply each input by a weight but then only the maximum and the minimum quantity of the products are summed together.

In formulas, if $v_1, v_2, \ldots$ are the inputs after being multiplied by their respective weights, the output $u$ of a MAM neuron is

$$u = \left[ \max_j v_j + \min_j v_j + b \right]^+ \tag{1}$$

where $b$ is the bias and $[\cdot]^+ = \max\{0, \cdot\}$ represents the nowadays common ReLU nonlinearity.

It is shown empirically that, starting from an architecture originally designed using MAC neurons, one may substitute them with MAM neurons in several hidden layers and use a proper training strategy to achieve the same performances as the corresponding MAC-only network. Yet, in the resulting hybrid network, one may leverage the extremely selective behaviour of min and max operations to reduce very aggressively the number of weights. MAM neurons can be pruned with almost every technique proposed in the literature with little to no modifications. As a motivating example, Table 1 reports some of the results described in [1] showing cases in which, once the quality level is set (in this case to 3% less accuracy than the original non-pruned network), MAM neuron substitution, retraining and pruning reduce the number of weights 1 to 2 orders of magnitude more than what is obtained by pruning the original MAC-only network. Moreover, these neurons can also be pruned iteratively requiring less training iterations to guarantee a given accuracy compared to standard MAC neurons.

Table 1: Approximate remaining interconnections in the hidden fully-connected layers with one-shot global magnitude pruning built either with MAC or MAM neurons.

|  | AlexNet + Cifar-10 | AlexNet + Cifar-100 | VGG-16 + ImageNet |
|---|---|---|---|
| Top-1 accuracy (3% lower than non-pruned network) | 87.69% | 63.89% | 61.03% |
| Surviving interconnections (MAC) | 1.01% | 25.01% | 10.82% |
| Surviving interconnections (MAM) | **0.06%** | **0.26%** | **0.04%** |

Though the equivalence between MAC-only and MAM&MAC networks has been demonstrated in practice, a change in the model of some neurons opens the problem of the abstract capability of such hybrid architectures. This contribution is a step forward in clarifying that, despite the locally different input-output relationships, also hybrid MAM&MAC networks enjoy some universal approximation capabilities analogous to those of the MAC-only networks.

## 1.1 Brief background on universal approximation properties

The development of models with universal approximation properties has been a significant breakthrough in many fields of science and engineering. In 1989 [3] proved that a network with a single hidden layer could approximate any continuous function, given enough hidden neurons. Some years later, [4] and [5] showed that also fuzzy systems could approximate any continuous function to arbitrary accuracy. These works were later extended to multiple inputs and outputs, demonstrating the universal approximation properties of fuzzy systems more broadly ([6, 7]). In the following years, a large number of researchers have studied the universal approximation properties of neural networks with MAC neurons in the case of bounded depth and arbitrary width ([8, 9]), bounded width and arbitrary depth ([10, 11, 12]) and bounded width and depth ([13, 14]). In the recent work [15], authors obtained the optimal minimum width bound of a neural network with arbitrary depth to retain universal approximation capabilities.

The research in this field is still very active and aims at proving the universal approximation capabilities of networks with different architectural or computational paradigm choices, such as deep convolutional neural networks [16], dropout neural networks [17], networks representing probability distributions [18] and spiking neural networks [19].

## 2 Mathematical model

We indicate with $\mathcal{L}(\cdot)$ a fully connected layer in which all neurons are based on the MAM paradigm (1). We consider networks with $N$ inputs collected in the vector $\boldsymbol{x} = (x_1, \dots, x_N)$, two MAM hidden layers producing a vector $\boldsymbol{z}(\boldsymbol{x}) = (z_1(\boldsymbol{x}), z_2(\boldsymbol{x}), \dots) = \mathcal{L}''\big(\mathcal{L}'(\boldsymbol{x})\big)$ and a single output $Z(\boldsymbol{x}) \in \mathbb{R}$ produced by a final layer that computes either the normalized linear combination

$$Z(\boldsymbol{x}) = \frac{\sum_k c_k z_k(\boldsymbol{x})}{\sum_k z_k(\boldsymbol{x})} \tag{2}$$

or the linear combination

$$Z(\boldsymbol{x}) = \sum_k c_k z_k(\boldsymbol{x}) \tag{3}$$

We normalize the input domain by assuming $x_i \in \mathbb{X} = [0, 1]$ for $i = 1, \dots, N$ and indicate with $\mathcal{Z}^*$ the family of functions in (2) while with $\mathcal{Z}$ the analogous family of functions in (3). Smoothness conditions on our target functions $f : \mathbb{X}^N \mapsto \mathbb{R}$ is formalized by assuming that they belong to $\mathcal{C}^d(\mathbb{X}^N)$, i.e., that their $d$-th order derivatives are continuous. Distances between functions are measured by means of the norms defined as

$$\|\phi\|_{k,p} = \left[ \int_{\mathbb{X}^N} |\phi(x)|^p \, \mathrm{d}x + k \sum_{j=1}^N \int_{\mathbb{X}^N} \left| \frac{\partial \phi}{x_j}(x) \right|^p \mathrm{d}x \right]^{1/p}$$

with $k = \{0, 1\}$ and $p \geq 1$.

## 3 Main results

Within the above framework, we prove two theorems that describe the universal approximation properties of DNNs using MAM neurons in the hidden layers.

**Theorem 1.** *For any function $f \in \mathcal{C}^0(\mathbb{X}^N)$ and any prescribed level of tolerance $\epsilon > 0$, there is a $Z \in \mathcal{Z}^*$ such that $\|f - Z\|_{0,\infty} \leq \epsilon$.*

**Theorem 2.** *For any function $f \in \mathcal{C}^2(\mathbb{X}^N)$, any prescribed level of tolerance $\epsilon > 0$ and finite $p \geq 1$, there is a $Z \in \mathcal{Z}$ such that $\|f - Z\|_{1,p} \leq \epsilon$.*

The proofs of both theorems are reported in Section 6 and are constructive. In particular, subnetworks in the cascade $\boldsymbol{z}(\boldsymbol{x}) = \mathcal{L}''\big(\mathcal{L}'(\boldsymbol{x})\big)$ are identified and programmed to make each $z_k(\boldsymbol{x})$ a weakly unimodal piecewise-linear function of the inputs, whose maximum is 1 and is reached in a hyper-rectangular subset of the domain, while the function vanishes for points far from the center of that hyper-rectangle. The shapes and positions of these functions can then be designed along with the values of the weights $c_k$ so that their combination by means of either (2) or (3) is capable of approximating the target function arbitrarily well as measured either by $\|\cdot\|_{1,p}$ or $\|\cdot\|_{0,\infty}$.

## 4 Examples

Figure 1 proposes a visual representation of the constructions behind Theorem 1 and Theorem 2 for $N = 2$. From left to right, we report the target function $f : \mathbb{X}^2 \to \mathbb{R}$

$$f(x_1, x_2) = \frac{(4x_1 - 2)(4x_2 - 2)\left(4x_1 + \frac{1}{2}\right)}{1 + (4x_1 - 2)^2 + (4x_2 - 2)^2} + 3 \tag{4}$$

and its approximation $Z \in \mathcal{Z}^*$ implied by the proof of Theorem 1 and its approximation $X \in \mathcal{Z}$ implied by the proof of Theorem 2. In both cases the parameter $n$ used in Section 6 is set to $n = 7$.

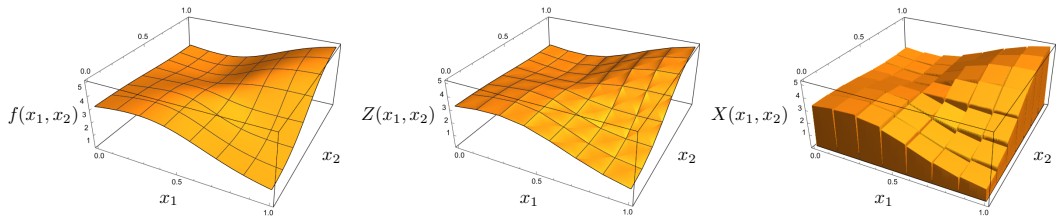

Figure 1: Three dimensional plot of a target function $f(x_1, x_2)$ and of its two approximations $Z(x_1, x_2) \in \mathcal{Z}^*$ implied by Theorem 1 and $X(x_1, x_2) \in \mathcal{Z}$ by Theorem 2.

## 5 Limitations

Theorem 1 and Theorem 2 rely on networks in which constraints are put neither on the layer width nor on the total number of neurons. Hence, despite proving universal approximation capabilities, they do not imply *efficient* approximation. Yet, such theoretical limitation is never strongly experienced in practice, since MAM networks are able to guarantee acceptable performance in real use cases. Nevertheless, a deeper look at universal approximation aimed at meeting efficiency will be the focus of future analysis.

## 6 Network construction and proofs of Theorems

### 6.1 Network construction

The aim of this subsection is to show that our network can be programmed to make the outputs of the second hidden layer specific weakly unimodal piecewise-linear functions $z_k(\boldsymbol{x})$ of the inputs.

**Lemma 1.** *Let $z$ be any of the outputs of the second hidden layer. For $N > 1$ and any choice of the quantities $\omega_1, \ldots, \omega_N \in [0, 1]$, $l_1, \ldots, l_N \geq 0$, $\delta_1^L, \ldots, \delta_N^L \geq 0$, and $\delta_1^R, \ldots, \delta_N^R \geq 0$, the two MAM hidden layers can be programmed to yield*

$$z(\boldsymbol{x}) = [1 - \Delta(\boldsymbol{x})]^+ \tag{5}$$

*where*

$$\Delta(\boldsymbol{x}) = \max_{i \in \{1, .., N\}} \left\{ 0, \frac{|x_i - \omega_i| - l_i}{\begin{cases} \delta_i^L & \text{if } x_i < \omega_i \\ \delta_i^R & \text{if } x_i \geq \omega_i \end{cases}} \right\} \tag{6}$$

*Proof of Lemma 1.* We assume that neurons in the first hidden layer come in pairs $(y_1^L, y_1^R, y_2^L, y_2^R, \ldots) = \mathcal{L}'(\boldsymbol{x})$ and the output of a pair depends on only one of the inputs.

Without any loss of generality, we assume that $y_i^L$ and $y_i^R$ depend only on $x_i$ for $i = 1, \ldots, N$ while all the other $N - 1$ input weights are set to $0$. The other outputs of the first hidden layer are involved in the computation of the outputs of the second hidden layer further to the $z$ we are considering.

For $y_i^L$ the non-null input weight is equal to $-1/\delta_i^L$ and the bias is $(\omega_i - l_i)/\delta_i^L$, while for $y_i^R$ the non-null input weight is equal to $1/\delta_i^R$ and bias is $(-\omega_i - l_i)/\delta_i^R$. By recalling (1) one gets

$$y_i^L = \left[ \frac{-x_i + \omega_i - l_i}{\delta_i^L} \right]^+ \quad \text{and} \quad y_i^R = \left[ \frac{x_i - \omega_i - l_i}{\delta_i^R} \right]^+ \tag{7}$$

In the second hidden layer, the neuron computing the $z$ we consider has all input weights equal to $0$ but those connecting to $y_1^L, y_1^R, \ldots, y_N^L, y_N^R$. Non-null input weights are equal to $-1$ and the bias is $1$ so that

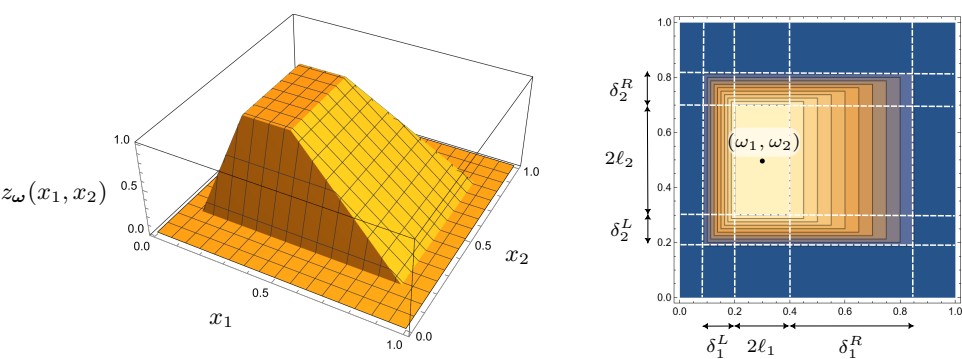

Figure 2: Three dimensional plot of a generic $z_{\boldsymbol{\omega}}(\boldsymbol{x})$ for $N = 2$ and its contour plot showing the role of the various parameters.

$$z = \left[\max_{i\in\{1,..,N\}}\{0, -y_i^{\mathrm{L}}, -y_i^{\mathrm{R}}\} + \min_{i\in\{1,..,N\}}\{0, -y_i^{\mathrm{L}}, -y_i^{\mathrm{R}}\} + 1\right]^{+} = \left[1 - \max_{i\in\{1,..,N\}}\{y_i^{\mathrm{L}}, y_i^{\mathrm{R}}\}\right]^{+} \quad (8)$$

Considering the last expression, note that, if $x_i \geq \omega_i$ then $y_i^{\mathrm{R}} \geq 0$ and $y_i^{\mathrm{L}} = 0$ while, if $x_i < \omega_i$ then $y_i^{\mathrm{R}} = 0$ and $y_i^{\mathrm{L}} \geq 0$. Hence, without loss of generality, we may assume that $x_i \geq \omega_i$ for $i = 1, \ldots, N$, being all other cases a variation of this one by suitable symmetry and scaling. With this, $y_i^{\mathrm{L}} = 0$ for $i = 1, \ldots, N$ and (8) becomes

$$z = \left[1 - \max_{i=1,\ldots,N}\left[\frac{x_i - \omega_i - l_i}{\delta_i^{\mathrm{R}}}\right]^{+}\right]^{+} = \left[1 - \max_{i=1,\ldots,N}\left\{0, \frac{x_i - \omega_i - l_i}{\delta_i^{\mathrm{R}}}\right\}\right]^{+} \quad (9)$$

that is equivalent to the thesis. $\square$

To interpret Lemma 1 note that $\Delta(\boldsymbol{x})$ is a scaled measure of how far the input vector $\boldsymbol{x}$ is from the hyper-rectangle centered at $\boldsymbol{\omega} = (\omega_1, \ldots, \omega_N)$ with sides $2l_1, \ldots, 2l_N$. Hence, $z(\boldsymbol{x})$ is maximum and equal to 1 if $\boldsymbol{x}$ belongs to such a hyper-rectangle and has a piecewise-linear decreasing profile when $\boldsymbol{x}$ gets further from $\boldsymbol{\omega}$. Figure 2 reports an example of a $z(\boldsymbol{x})$ when $N = 2$.

In the following, we will assume that each neuron in the second hidden layer matches a whole subnetwork as implied by Lemma 1. With this, we may re-index the outputs of the second hidden layer as $z_{\boldsymbol{\omega}}(\boldsymbol{x})$ associating each of them with the center of the hyper-rectangle in which $z_{\boldsymbol{\omega}}(\boldsymbol{x}) = 1$. The same is done with the corresponding weights $c_{\boldsymbol{\omega}}$ in the output layers.

## 6.2 Universal approximation properties with normalized linear output neuron

Given a positive integer $n$, define $\Omega = \left\{0, \frac{1}{n}, \frac{2}{n}, \ldots, 1\right\}^N$ and include in the two hidden layers all the subnetworks implied by Lemma 1 to implement the function $z_{\boldsymbol{\omega}}(\boldsymbol{x})$ for each $\boldsymbol{\omega} \in \Omega$.

In each of these subnetworks set $\delta_i^{\mathrm{L}} = \delta_i^{\mathrm{R}} = \delta = 1/n$ for $i = 1, \ldots, N$ and $l_i = 0$ for $i = 1, \ldots, N$.

With this, $z_{\boldsymbol{\omega}}(\boldsymbol{x})$ is and $(N+1)$-dimensional pyramid whose base is an $N$-dimensional hypercube with sides of length $2\delta$ and center in $\boldsymbol{\omega}$.

*Proof of Theorem 1.* Note first that for any given $\boldsymbol{x} \in \mathbb{X}^N$, only a limited number of functions $z_{\boldsymbol{\omega}}(\boldsymbol{x})$ are not null. In particular, if $k_i = \lfloor nx_i \rfloor$ for $i = 1, \ldots, N$ is the largest integer not exceeding $nx_i$, then $z_{\boldsymbol{\omega}}(\boldsymbol{x}) > 0$ only if $\boldsymbol{\omega}$ belongs to the set $\Omega_{\boldsymbol{x}} = \{k_1\delta, (k_1 + 1)\delta\} \times \cdots \times \{k_N\delta, (k_N + 1)\delta\}$ that contains the $2^N$ corners of the $N$-dimensional hypercube $C_{\boldsymbol{x}} = [k_1\delta, (k_1 + 1)\delta] \times \cdots \times [k_N\delta, (k_N + 1)\delta]$. Hence, we may evaluate $Z(\boldsymbol{x})$ focusing on functions $z_{\boldsymbol{\omega}}(\boldsymbol{x})$ with $\boldsymbol{\omega} \in \Omega_{\boldsymbol{x}}$.

Define the functions

$$\zeta_{\boldsymbol{\omega}}(x) = \frac{z_{\boldsymbol{\omega}}(\boldsymbol{x})}{\sum_{\boldsymbol{\omega}' \in \Omega} z_{\boldsymbol{\omega}'}(\boldsymbol{x})} \tag{10}$$

that are such that $\sum_{\boldsymbol{\omega} \in \Omega} \zeta_{\boldsymbol{\omega}}(\boldsymbol{x}) = \sum_{\boldsymbol{\omega} \in \Omega_{\boldsymbol{x}}} \zeta_{\boldsymbol{\omega}}(\boldsymbol{x}) = 1$ for any $\boldsymbol{x} \in \mathbb{X}^N$, and set $c_{\boldsymbol{\omega}} = f(\boldsymbol{\omega})$ for each $\boldsymbol{\omega} \in \Omega$.

The error $\|f(\boldsymbol{x}) - Z(\boldsymbol{x})\|_{0,\infty}$ in Theorem 1 can be written as

$$\left\| f(\boldsymbol{x}) - \sum_{\boldsymbol{\omega} \in \Omega_{\boldsymbol{x}}} f(\boldsymbol{\omega}) \zeta_{\boldsymbol{\omega}}(\boldsymbol{x}) \right\|_{0,\infty} = \left\| \sum_{\boldsymbol{\omega} \in \Omega_{\boldsymbol{x}}} [f(\boldsymbol{x}) - f(\boldsymbol{\omega})] \zeta_{\boldsymbol{\omega}}(\boldsymbol{x}) \right\|_{0,\infty} \le \max_{\boldsymbol{x} \in \mathbb{X}^N} \max_{\substack{\boldsymbol{\xi} \in C_{\boldsymbol{x}} \\ \boldsymbol{\omega} \in \Omega_{\boldsymbol{x}}}} |f(\boldsymbol{\xi}) - f(\boldsymbol{\omega})|$$

Since $f : \mathbb{X}^N \mapsto \mathbb{R}$ is continuous on the compact domain $\mathbb{X}^N$, it is also uniformly continuous and, for any given level of tolerance $\epsilon > 0$, there is a $\Delta x$ such that for any $\boldsymbol{x}', \boldsymbol{x}'' \in \mathbb{X}^N$ with distance $\|\boldsymbol{x}' - \boldsymbol{x}''\|_2 \le \Delta x$ we have $|f(\boldsymbol{x}') - f(\boldsymbol{x}'')| \le \epsilon$. For a given $\boldsymbol{x}$, the distance between any $\boldsymbol{\xi} \in C_{\boldsymbol{x}}$ and any $\boldsymbol{\omega} \in \Omega_{\boldsymbol{x}}$ is $\|\boldsymbol{\xi} - \boldsymbol{\omega}\|_2 \le \delta\sqrt{N}$. Since $\delta = 1/n$ we can select $n$ so that

$$\|f(\boldsymbol{x}) - Z(\boldsymbol{x})\|_{0,\infty} \le \max_{\boldsymbol{x} \in \mathbb{X}^N} \max_{\substack{\boldsymbol{\xi} \in C_{\boldsymbol{x}} \\ \boldsymbol{\omega} \in \Omega_{\boldsymbol{x}}}} |f(\boldsymbol{\xi}) - f(\boldsymbol{\omega})| \le \epsilon$$

$\square$

## 6.3 Universal approximation properties with linear output neuron

In this case, the approximation capabilities of our network over the whole domain depend on the local behaviour of subnetworks converging not in a single second-hidden-layer neuron but in $2N$ second-hidden-layer neurons.

Formally, given a center $\boldsymbol{\omega} \in \mathbb{X}^N$ we include in a subnetwork neurons of the second hidden layer with outputs labelled $z_{\boldsymbol{\omega}^{1-}}, z_{\boldsymbol{\omega}^{1+}}, \ldots, z_{\boldsymbol{\omega}^{N-}}, z_{\boldsymbol{\omega}^{N+}}$ as well as all the previous neurons needed to compute such outputs.

The expression of each $z_{\boldsymbol{\omega}^{j\pm}}$ is given by Lemma 1 and thus is defined by the center point $\boldsymbol{\omega}^{j\pm} = (\omega_1^{j\pm}, \ldots, \omega_N^{j\pm})$, by the slopes $\delta_1^{\text{L},j\pm}, \ldots, \delta_N^{\text{L},j\pm}$ and $\delta_1^{\text{R},j\pm}, \ldots, \delta_N^{\text{R},j\pm}$, as well as by the side lengths $l_1^{j\pm}, \ldots, l_N^{j\pm}$.

In a subnetwork, everything depends on two quantities $\delta, \ell \ge 0$ that are used to set

$$\omega_i^{j\pm} = \begin{cases} \omega_i & \text{if } i \ne j \\ \omega_i \pm \ell & \text{if } i = j \end{cases} \qquad l_i^{j\pm} = \begin{cases} \ell & \text{if } i \ne j \\ 0 & \text{if } i = j \end{cases}$$

$$\delta_i^{\text{R},j-} = \delta \qquad\qquad \delta_i^{\text{R},j+} = \begin{cases} \delta & \text{if } i \ne j \\ 2\ell & \text{if } i = j \end{cases}$$

$$\delta_i^{\text{L},j-} = \begin{cases} \delta & \text{if } i \ne j \\ 2\ell & \text{if } i = j \end{cases} \qquad \delta_i^{\text{L},j+} = \delta$$

for $i, j = 1, \ldots, N$.

To give some intuitive grounding to the above definitions, Figure 3 reports example profiles for 4 output functions $z_{\boldsymbol{\omega}^{1-}}, z_{\boldsymbol{\omega}^{1+}}, z_{\boldsymbol{\omega}^{2-}}, z_{\boldsymbol{\omega}^{2+}}$ with $N = 2$.

Given a center $\boldsymbol{\omega}$, the same quantities $\delta$ and $\ell$ allow to define the two domain subsets

$$X_{\boldsymbol{\omega}}^{\blacksquare} = \left\{ \boldsymbol{x} \in \mathbb{X}^N \ \middle| \ \max_{i=1,\ldots,N} \{|x_i - \omega_i|\} \le \ell \right\} \qquad X_{\boldsymbol{\omega}}^{\square} = \left\{ \boldsymbol{x} \in \mathbb{X}^N \ \middle| \ \ell < \max_{i=1,\ldots,N} \{|x_i - \bar{\omega}_i|\} \le \ell + \delta \right\}$$

as well as $X_{\boldsymbol{\omega}} = X_{\boldsymbol{\omega}}^{\blacksquare} \cup X_{\boldsymbol{\omega}}^{\square}$.

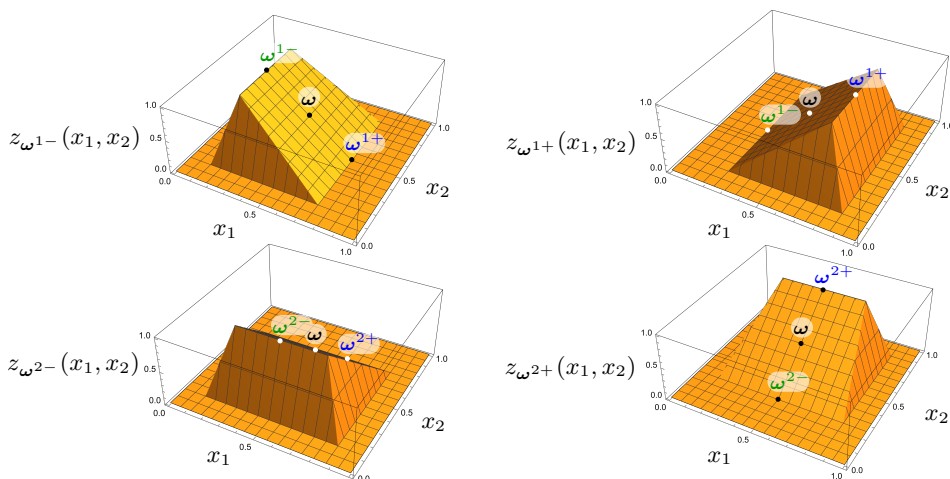

Figure 3: Three dimensional plots of the functions $z_{\boldsymbol{\omega}^{1-}}, z_{\boldsymbol{\omega}^{1+}}, z_{\boldsymbol{\omega}^{2-}}, z_{\boldsymbol{\omega}^{2+}}$ with $N = 2$.

The approximation capabilities depend on the behaviour of the output of the subnetworks in the three disjoint domains $X_{\boldsymbol{\omega}}^{\blacksquare}$, $X_{\boldsymbol{\omega}}^{\square}$, and $\mathbb{X}^N \setminus X_{\boldsymbol{\omega}}$.

It is easy to see that if $\boldsymbol{x} \in \mathbb{X}^N \setminus X_{\boldsymbol{\omega}}$ then $z_{\boldsymbol{\omega}^{j\pm}} = 0$ for $j = 1, \ldots, N$.

For $\boldsymbol{x} \in X_{\boldsymbol{\omega}}^{\blacksquare}$ the following Lemma holds.

**Lemma 2.** *Given any choice of $N + 1$ coefficients $a$ and $b_j$ for $j = 1, \ldots, N$, one may choose $2N$ weights $c^{j\pm}$ with $j = 1, \ldots, N$ such that*

$$Z_{\boldsymbol{\omega}}\left(\boldsymbol{x}\right) \equiv \sum_{j=1}^{N} c^{j\pm} z_{\boldsymbol{\omega}^{j\pm}}\left(\boldsymbol{x}\right) = a + \sum_{j=1}^{N} b_j x_j \tag{11}$$

*for any $\boldsymbol{x} \in X_{\boldsymbol{\omega}}^{\blacksquare}$, where $Z_{\boldsymbol{\omega}}\left(\boldsymbol{x}\right)$ remains implicitly defined.*

*Proof of Lemma 2.* Due to the definition of $\boldsymbol{\omega}^{j\pm}$ we have

$$X_{\boldsymbol{\omega}}^{\blacksquare} = [\omega_1 - \ell, \omega_1 + \ell] \times \cdots \times [\omega_N - \ell, \omega_N + \ell] = [\omega_1^{1-}, \omega_1^{1+}] \times \cdots \times [\omega_N^{N-}, \omega_N^{N+}]$$

Hence, if $\boldsymbol{x} \in X_{\boldsymbol{\omega}}^{\blacksquare}$ we know that $\omega_j^{j-} \le x_j \le \omega_j^{j+}$ for $j = 1, \ldots, N$.

Moreover, since by definition for any $i, j = 1, \ldots, N$ and $i \ne j$ we have $\omega_i^{j+} - \omega_i^{j-} = 2\ell$ and $\omega_i^{j-} + \omega_i^{j+} = 2\omega_i$, then $\left|x_i - \omega_i^{j\pm}\right| \le \ell$ when $i \ne j$. Therefore, one can apply Lemma 1 and compute $\Delta(\boldsymbol{x})$, for which all the terms in (6) but $\left|x_j - \omega_j^{j\pm}\right|$ are non-positive, thus yielding $z_{\omega^{j\pm}}\left(\boldsymbol{x}\right) = 1 - \left|x_j - \omega_j^{j\pm}\right|/(2\ell)$.

Without any loss of generality, translate $X_{\boldsymbol{\omega}}$ so that $\boldsymbol{\omega} = (\ell, \ldots, \ell)$. This implies $\omega_j^{j-} = 0$ and $\omega_j^{j+} = 2\ell$ for $j = 1, \ldots, N$, thus yielding $z_{\boldsymbol{\omega}^{j-}}\left(\boldsymbol{x}\right) = 1 - \frac{x_j}{2\ell}$ and $z_{\boldsymbol{\omega}^{j+}}\left(\boldsymbol{x}\right) = \frac{x_j}{2\ell}$. With this,

$$\sum_{j=1}^{N} c^{j\pm} z_{\boldsymbol{\omega}^{j\pm}}\left(\boldsymbol{x}\right) = \sum_{j=1}^{N} \left[c^{j-}\left(1 - \frac{x_j}{2\ell}\right) + c^{j+}\frac{x_j}{2\ell}\right] = \sum_{j=1}^{N} c^{j-} + \sum_{j=1}^{N} \left(c^{j+} - c^{j-}\right)\frac{x_j}{2\ell}$$

that can yield any affine function $f(x) = a + \sum_{j=1}^{N} b_j x_j$ by setting, for $j = 1, \ldots, N$,

$$c^{j-} = \frac{a}{N} \quad \text{and} \quad c^{j+} = c^{j-} + 2\ell b_j \tag{12}$$

$\square$

Finally, what happens for $\boldsymbol{x} \in X_{\boldsymbol{\omega}}^{\square}$ is described by the following Lemma.

**Lemma 3.** *If the $2N$ weights $c^{j\pm}$ with $j = 1, \ldots, N$ are set according to Lemma 2 so that $Z_{\boldsymbol{\omega}}\left(\boldsymbol{x}\right) = a + \sum_{j=1}^{N} b_j x_j$ for any $x \in X_{\boldsymbol{\omega}}^{\blacksquare}$, for coefficients satisfying $|a|, |b_j| \le M$ for some $M > 0$ and $j = 1, \ldots, N$, then $\left|Z_{\boldsymbol{\omega}}\left(\boldsymbol{x}\right)\right| \le 3MN$ for any $\boldsymbol{x} \in X_{\boldsymbol{\omega}}$ and thus for any $\boldsymbol{x} \in X_{\boldsymbol{\omega}}^{\square}$.*

*Proof of Lemma 3.* From $|a|, |b_j| \le M$ and from (12) we get $\left| c^{j-} \right| \le M/N$ and $\left| c^{j+} \right| \le M/N + 2\ell M$.

Overall, since $\ell \le 1$ and $N \ge 1$ we have $\left| c^{j\pm} \right| \le 3M$. Since $0 \le z_{\boldsymbol{\omega}^{j\pm}} \le 1$ and $Z_{\boldsymbol{\omega}}(\boldsymbol{x}) = \sum_{j=1}^{N} c^{j\pm} z_{\boldsymbol{\omega}^{j\pm}}(\boldsymbol{x})$ we finally get the thesis. $\qquad\square$

The above characterization of the output of $Z$-subnetworks allows to prove their local approximation capabilities.

**Lemma 4.** *Given any function* $f \in \mathcal{C}^2 \left( \mathbb{X}^N \right)$, *there are two constants* $P, Q > 0$ *such that*

$$
\begin{aligned}
E_{\boldsymbol{\omega}} &\equiv \int_{X_{\boldsymbol{\omega}}} |f(\boldsymbol{x}) - Z_{\boldsymbol{\omega}}(\boldsymbol{x})|^p \, \mathrm{d}\boldsymbol{x} + \sum_{j=1}^{N} \int_{X_{\boldsymbol{\omega}}} \left| \frac{\partial f}{\partial x_j}(\boldsymbol{x}) - \frac{\partial Z_{\boldsymbol{\omega}}}{\partial x_j}(\boldsymbol{x}) \right|^p \mathrm{d}\boldsymbol{x} \\
&\le (2\ell + 2\delta)^N \left\{ P\ell^p \left[ 1 - o\left( \delta/\ell \right) \right] + Qo\left( \delta/\ell \right) \right\}
\end{aligned}
$$

*with* $o(\cdot) = 1 - 1/(1+\cdot)^N$

*Proof of Lemma 4.* Since $f \in \mathcal{C}^2 \left( \mathbb{X}^N \right)$ and $\mathbb{X}^N$ is compact, $M_0, M_1, M_2 \ge 0$ exists such that

$$
|f(\boldsymbol{x})| \le M_0, \qquad \left| \frac{\partial f}{\partial x_i}(\boldsymbol{x}) \right| \le M_1, \qquad \left| \frac{\partial^2 f}{\partial x_i x_j}(\boldsymbol{x}) \right| \le M_2 \tag{13}
$$

for any $\boldsymbol{x} \in \mathbb{X}^M$ and $i, j = 1, \ldots, N$.

Assuming $\boldsymbol{x} \in X_{\boldsymbol{\omega}}^{\blacksquare}$, and thus $|x_i - \omega_i| \le \ell$, the above bounds can be used jointly with the Taylor expansions of $f$ and its derivatives around $\boldsymbol{\omega}$

$$
f(\boldsymbol{x}) = f(\boldsymbol{\omega}) + \sum_{i=1}^{N} \frac{\partial f}{\partial x_i}(\boldsymbol{\omega})(x_i - \omega_i) + \sum_{i=1}^{N} \sum_{j=1}^{N} R_{i,j}(\boldsymbol{x})(x_i - \omega_i)(x_j - \omega_j) \tag{14}
$$

$$
\frac{\partial f}{\partial x_i}(\boldsymbol{x}) = \frac{\partial f}{\partial x_i}(\boldsymbol{\omega}) + \sum_{j=1}^{N} S_{i,j}(\boldsymbol{x})(x_j - \omega_j) \qquad i = 1, \ldots, N \tag{15}
$$

to ensure that their error terms satisfy

$$
\left| \sum_{i=1}^{N} \sum_{j=1}^{N} R_{i,j}(\boldsymbol{x})(x_i - \omega_i)(x_j - \omega_j) \right| \le N^2 \ell^2 \frac{1}{2} \max_{k,l=1,\ldots,N} \max_{\boldsymbol{\xi} \in \mathbb{X}^N} \left| \frac{\partial^2 f}{\partial x_k x_l}(\boldsymbol{\xi}) \right| \le \frac{1}{2} M_2 N^2 \ell^2 \tag{16}
$$

and

$$
\left| \sum_{j=1}^{N} S_{i,j}(\boldsymbol{x})(x_j - \omega_j) \right| \le N^2 \ell^2 \frac{1}{2} \max_{j=1,\ldots,N} \max_{\boldsymbol{\xi} \in \mathbb{X}^N} \left| \frac{\partial^2 f}{\partial x_i x_j}(\boldsymbol{\xi}) \right| \le \frac{1}{2} M_2 N \ell \qquad i = 1, \ldots, N \tag{17}
$$

Again focusing on $\boldsymbol{x} \in X_{\boldsymbol{\omega}}^{\blacksquare}$, exploit Lemma 2 to set the weights $c^{j\pm}$ yielding

$$
Z_{\boldsymbol{\omega}}(\boldsymbol{x}) = f(\boldsymbol{\omega}) + \sum_{i=1}^{N} \frac{\partial f}{\partial x_i}(\boldsymbol{\omega})(x_i - \omega_i) = \left[ f(\boldsymbol{\omega}) - \sum_{i=1}^{N} \frac{\partial f}{\partial x_i}(\boldsymbol{\omega}) \omega_i \right] + \sum_{i=1}^{N} \frac{\partial f}{\partial x_i}(\boldsymbol{\omega}) x_i \tag{18}
$$

which is also such that $\frac{\partial Z_{\boldsymbol{\omega}}}{\partial x_i}(\boldsymbol{x}) = \frac{\partial f}{\partial x_u}(\boldsymbol{\omega})$.

Hence, we may program $Z_{\boldsymbol{\omega}}$ to reproduce the beaviour of $f$ and its derivatives in $X_{\boldsymbol{\omega}}^{\blacksquare}$, and the approximation errors can be derived exploiting (14) with (16) and (15) with (17) to obtain

$$
|Z_{\boldsymbol{\omega}}(\boldsymbol{x}) - f(\boldsymbol{x})| \le \frac{1}{2} M_2 N^2 \ell^2, \qquad \left| \frac{\partial Z_{\boldsymbol{\omega}}}{\partial x_i}(\boldsymbol{x}) - \frac{\partial f}{\partial x_i}(\boldsymbol{x}) \right| \le \frac{1}{2} M_2 N \ell \tag{19}
$$

To address the case $\boldsymbol{x} \in X_{\boldsymbol{\omega}}^{\square}$, we may apply Lemma 3. By matching (18) with (13) we get that $|a| \le M_0 + M_1 N$ and $|b_i| \le M_1 \le M_0 + M_1 N$ for $i = 1, \ldots, N$. Hence, if $x \in X_{\omega}^{\square}$, then if $M_3 = M_0(1 + 3N) + 3M_1 N^2$ we have

$$|Z_{\boldsymbol{\omega}}(\boldsymbol{x}) - f(\boldsymbol{x})| \le M_3, \qquad \left|\frac{\partial Z_{\boldsymbol{\omega}}}{\partial x_i}(\boldsymbol{x}) - \frac{\partial f}{\partial x_i}(\boldsymbol{x})\right| = \left|\frac{\partial f}{\partial x_i}(\boldsymbol{\omega}) - \frac{\partial f}{\partial x_i}(\boldsymbol{x})\right| \le 2M_1 \qquad (20)$$

Since we have different error bounds in $X_{\boldsymbol{\omega}}^{\blacksquare}$ and $X_{\hat{\omega}}^{\square}$, we bound the overall error $E_{\boldsymbol{\omega}}$ by splitting

$$
\begin{aligned}
E_{\boldsymbol{\omega}} \quad = \quad & \int_{X_{\boldsymbol{\omega}}^{\blacksquare}} |f(\boldsymbol{x}) - Z_{\boldsymbol{\omega}}(\boldsymbol{x})|^p \, \mathrm{d}\boldsymbol{x} + \sum_{j=1}^{N} \int_{X_{\boldsymbol{\omega}}^{\blacksquare}} \left|\frac{\partial f}{\partial x_j}(\boldsymbol{x}) - \frac{\partial Z_{\boldsymbol{\omega}}}{\partial x_j}(\boldsymbol{x})\right|^p \, \mathrm{d}\boldsymbol{x} + \\
& \int_{X_{\boldsymbol{\omega}}^{\square}} |f(\boldsymbol{x}) - Z_{\boldsymbol{\omega}}(\boldsymbol{x})|^p \, \mathrm{d}\boldsymbol{x} + \sum_{j=1}^{N} \int_{X_{\boldsymbol{\omega}}^{\square}} \left|\frac{\partial f}{\partial x_j}(\boldsymbol{x}) - \frac{\partial Z_{\boldsymbol{\omega}}}{\partial x_j}(\boldsymbol{x})\right|^p \, \mathrm{d}\boldsymbol{x} +
\end{aligned}
$$

and apply (19) and (20) to bound each integrand. Adding the fact that the measure of $X_{\omega}^{\blacksquare}$ is $(2\ell)^N$, while the measure of $X_{\omega}^{\square}$ is $(2\ell + 2\delta)^N - (2\ell)^N$ we obtain

$$E_{\boldsymbol{\omega}} \le \left[\left(\frac{1}{2} M_2 N^2 \ell^2\right)^p + \left(\frac{1}{2} M_2 N \ell\right)^p\right] (2\ell)^N + \left[M_3^p + (2M_1)^p\right] \left[(2\ell + 2\delta)^N - (2\ell)^N\right]$$

from which we may set $P = \left(\frac{1}{2} M_2 N^2\right)^p + \left(\frac{1}{2} M_2 N\right)^p$ and $Q = M_3^p + (2M_1)^p$ to get the thesis. $\square$

We are now in the position of proving our second result.

*Proof of Theorem 2.* For $n > 0$ integer define $\delta$ and $\ell$ such that $\delta = \ell^2$ and $2\ell + 2\delta = 1/n$. Let also $\Omega = \left\{\frac{1}{2n}, \frac{3}{2n}, \ldots, \frac{2n-1}{2n}\right\}^N$ so that $\mathbb{X}^N$ is partitioned in $n^N$ hyper-cubes $X_{\boldsymbol{\omega}}$ with centers $\boldsymbol{\omega} \in \Omega$ and side $2\ell + 2\delta$. The output of the whole network is $Z(\boldsymbol{x}) = \sum_{\boldsymbol{\omega} \in \Omega} Z_{\boldsymbol{\omega}}(\boldsymbol{x})$.

Since $Z_{\boldsymbol{\omega}}(\boldsymbol{x})$ is null for $\boldsymbol{x} \notin X_{\boldsymbol{\omega}}$, the error measure over $\mathbb{X}^N$ can be decomposed into

$$\|f - Z\|_{1,p}^p = \sum_{\boldsymbol{\omega} \in \Omega} \left\{\int_{X_{\boldsymbol{\omega}}} |f(\boldsymbol{x}) - Z_{\boldsymbol{\omega}}(\boldsymbol{x})|^p \, \mathrm{d}\boldsymbol{x} + \sum_{j=1}^{N} \int_{X_{\boldsymbol{\omega}}} \left|\frac{\partial f}{\partial x_j}(\boldsymbol{x}) - \frac{\partial Z_{\boldsymbol{\omega}}}{\partial x_j}(\boldsymbol{x})\right|^p \, \mathrm{d}\boldsymbol{x}\right\}$$

Each of the terms in the last sum can be bounded using Lemma 4 in which we may also substitute $2\ell + 2\delta = 1/n$ and $\delta = \ell^2$ to yield

$$\|f - Z\|_{1,p}^p \le \sum_{\boldsymbol{\omega} \in \Omega} \frac{1}{n^N} \left\{P\ell^p \left[1 - o(\ell)\right] + Qo(\ell)\right\} = P\ell^p \left[1 - o(\ell)\right] + Qo(\ell)$$

Since when $n \to \infty$ we have $\ell \to 0$ and thus $o(\ell) \to 0$ the thesis is proven. $\square$

## 7 Conclusions

We established that neural networks in which hidden MAC neurons are substituted with MAM neurons to obtain more aggressively prunable architectures are still universal approximators.

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
