# OpenReview forum: "On the Universal Approximation Properties of Deep Neural Networks using MAM Neurons"
_NeurIPS.cc/2023/Conference — Submitted to NeurIPS 2023_

### Official Review · Reviewer_p1mV · 2023-06-28

**Soundness:** 2 fair
**Presentation:** 2 fair
**Contribution:** 1 poor
**Rating:** 3
**Confidence:** 5

**Summary:**

This manuscript proves universal approximation results for MAM neurons. MAM neurons are essentially ReLU neurons that operate on the sum of the maximum and the minimum of the weighted inputs, plus a bias. Previous work claims these neurons are useful for reducing the memory footprint of deep neural networks. Thus, in short, the paper proves how any real-valued continuous function defined on a compact set can be approximated to any arbitrary degree of precision by a network consisting primarily of MAM neurons (for instance in the sup norm).

**Strengths:**

The paper is relatively well-written and easy to understand. Universal approximation properties are important, and proving them is a key step when new activation functions are introduced.

**Weaknesses:**

There are two problems with this contribution. First, there are many universal approximation results in the literature already, and thus this contribution is perceived as incremental. Second, and more importantly, an even better result can easily be proved by adapting well-known techniques for providing simple proofs of universal approximation properties (the authors do not seem to be aware of the existence of such techniques).One only needs to slightly tweak the proof that is used for the case of ReLU neurons.   In particular, it is easy to show that any continuous function from [0,1] to R, can be approximated by a neural network with a single linear output unit and two hidden layers of MAM neurons to any degree of precision \epsilon. To see this, note that f is *uniformly* continuous over [0,1]. Thus the [0,1] interval can be subdivided into small intervals of size \alpha, such that within any such interval f is contained "within a sleeve of thickness \epsilon". Now connect the input to all the MAM neurons in the first hidden layer with identical weights equal to 0.5. As a result, the max + min portion of the input of each MAM neuron in the first hidden layer is equal to the input x. Now select a sequence of (arithmetically) increasing biases so that:
1) the first hidden neuron is turned on if x in the first interval of size \alpha and all the other hidden neurons produce a zero;
2) the first and second neurons are turned on if x is in the second interval of size \alpha and all other hidden neurons produce a zero;
etc. In other words, essentially code the value of x by the number of neurons that are turned on in the hidden layer. It is then easy to see how to design the following hidden layer and connect it to the single linear output neuron to obtain the desired approximation.

**Questions:**

The authors may consider providing a simpler proof, along the lines sketched above, for a potential submission to a lesser conference.

**Limitations:**

The main limitation of the universal approximation results is that the hidden layers can be arbitrary large (depending on the size of \alpha in the proof sketched above). And thus in general the constructive proofs of these results are not practical. This is a well known limitation and the authors, to their credit, do mention it.

---

> ### Author Rebuttal · Authors · 2023-08-09
>
> We thank the reviewer for their time and their valuable feedback. While we are naturally disappointed by the outcome, we value the reviewer's expertise and the insights they provided. We answer here to the reviewer's comments, hoping they could change their idea about this work.
>
> Actually, we are aware of the literature on universal approximation by means of neural networks with ReLU activations, but we had to resort to a different approach to cope with the double requirement of i) considering multi-input networks ii) using MAM neurons in all layers but the last one. These are, in fact, the conditions presented in reference [1] of the paper in which one is able to leverage the features of the MAM neurons.
>
> Hence, the rules of our approximation game prevent the use of linear combination in all layers but the last one. Regrettably, the classical multi-input universal approximation results that are general enough to include ReLU nonlinearities, basically hinge on the model described, for example, in
> [Hornik "Approximation Capabilities of Multilayer Feedforward Networks", Neural Networks, 1992], i.e., on an input-output relationship of the kind
>
> $$
> \sum_j y_j \Psi\left(\left\langle a_j,x\right\rangle+b_j \right)
> $$
>
> where $x$ is the input vector, $\Psi$ is the non-linearity, $y_j$ are the coefficients of the output linear layer, $a_j$ is the $j$-th coefficient vector, $b_j$ is the $j$-th bias and $\left\langle \cdot,\cdot\right\rangle$ stands for the scalar product.
>
> MAC+ReLU Networks fit within this model if one aggregates, for example, three properly programmed ReLU to generate the "bounded and non-constant" function for which universal approximation can be proved (and it is easy to see that this aggregation of multiple ReLU profiles to yield a trapezoidal shape is exactly what we do in our first layer).
>
> Regrettably, if one wants to use MAM neurons in all but the last layer, then the scalar product cannot be used, and this spoils Hornik-like general results.
>
> Clearly, this is not perceived if sticking to only one input, as in the example given by the reviewer, which is indeed correct but cannot be easily generalized.
>
> In fact, even adopting the non-local strategy suggested by the reviewer, when going from one input to multiple inputs, one needs to provide to the final linear combination some building blocks that depend simultaneously on all the inputs.
>
> As the scalar product is out of the reach of a MAM neuron, this must be done *after* profiles depending separately on each input are computed, and thus in a subsequent layer.
>
> This is why a second layer is needed, and would be also needed when upscaling to multiple inputs the simple approach proposed by the reviewer if no pre-processing by a scalar product is allowed.
>
> This said, aggregating single input profiles into multi-input profiles using only MAM neurons (no MAC neurons allowed until the last layer) is not straightforward and, even if it were possible starting from the profiles suggested by the reviewers, the complexity of the second layer would be analogous.
>
> Overall, though tempting, the approach put forward by the reviewer loses its simplicity when tackling the multi-input case without being able to resort to a pre-processing of the input by means of a scalar product that is not available in our model.
>
> For these reasons, though we appreciate the clever suggestion from the reviewer we still think that the complexity we have highlighted and tackled in the paper cannot be easily circumvented and that implies the contribution cannot be seen as incremental.

---

> > ### Comment · Reviewer_p1mV · 2023-08-17
> >
> > I thank the authors for their response but I continue to view this contribution as incremental and thus leave my score unchanged.

---

> > > ### Author Response · Authors · 2023-08-20
> > >
> > > Though we cannot but thank the reviewer for the additional time spent on our paper, we would like to comment on the reply we received.
> > >
> > > The reviewer seems to think that our results can be derived somehow easily from existing theory and offered a one-input example that may seem to substantiate this.
> > >
> > > Yet, our rebuttal explains that the straighfroward construction proposed by the reviewer cannot be extended to more than one input and thus is of no help in proving our results in a simpler way.
> > >
> > > The explanation starts from the classical tool on which many universal approximation results are based and shows that one the required features (a linear combination of the neuron inputs *before* the nonlinearity) is not present in MAM architectures.
> > >
> > >
> > > Since the reviewer does not reply on this point we assume that the issue is cleared and thus it is recognized that classical universal approximation theorems cannot be extended in a straightforward way to prove our Theorem 1 and Theorem 2.
> > >
> > >
> > > Add to all this that our Theorem 2 provides universal approximation of differential features of the target that is commonly a separate result.
> > >
> > >
> > > In the light of all this we do believe that labeling our result as incremental is technically wrong and should not be accepted as the basis for a decision.

---

### Official Review · Reviewer_Ar7s · 2023-07-03

**Soundness:** 2 fair
**Presentation:** 1 poor
**Contribution:** 2 fair
**Rating:** 3
**Confidence:** 4

**Summary:**

This paper demonstrates that the network can still maintain the universal approximation property after substituting the classical MAC hidden neurons of neural networks with the MAM neurons, which only rely on the maximum and minimum elements of the summation, allowing for more aggressive pruning. Specifically, the authors consider a network with two hidden MAM layers and two kinds of output layers. They show that the networks can achieve universal approximation capabilities under different norms for target functions with varying smoothness. The constructive proof of the first case utilizes a similar idea to the partition of unity, and the second one decomposes the whole domain and deals with the local behavior of the subnetworks involving the second hidden layer.

**Strengths:**

The authors provide a universal approximation result for a recently proposed hidden neuron for neural networks with two different output designs. The constructive proof introduces localized hyper-rectangles, which may inspire other step-function-like construction in some approximation problems. It’s also helpful to provide some intuitive illustrations of the construction.

**Weaknesses:**

While the result is somewhat interesting, it fails to provide more compelling evidence for the newly proposed MAM neurons. In the introduction, MAM neurons’ main advantage is that they can allow more aggressive pruning. But the theorems and the proof process seem to have no investigation about these properties. And there is no further discussion about the academic and practical potentials of MAM neurons, weakening its significance and attractiveness as well as this work.

In Section 5, the authors claim that the theorem has no constraints on the layer width, which contradicts the conventional universal approximation properties. However, the proof introduces a parameter n that needs to be chosen sufficiently large (See line 158). Interestingly, this parameter seems to be related to the width or scale of the neural network according to its definition in line 141. Hence, there appears to be a potential contradiction in this context.

There are several evident traces of incompleteness throughout the manuscript, such as lines 141-145, equation after line 170 and line 216, and the unfinished Conclusions section in lines 229-230, indicating that this paper was hastily written and has not undergone thorough revisions.

The inadequate mathematical formatting in the manuscript has resulted in difficulties in comprehending the proof. Some mathematical notations used in this paper contradict commonly used notations in classical theories. The LaTeX formatting in the manuscript is not standardized.

**Questions:**

One critical issue with the inadequate mathematical formatting is that there is a lack of notation related to the width of the hidden layers in the network, resulting in significant ambiguity regarding the lengths of vectors and the ranges of summation throughout the manuscript, especially in the proof of Lemma 1.

Some mathematical notations may need to be changed, such as the norm defined after line 81, which might be easily misread as the Sobolev norm, the L’(x) and L’’(x) in line 74 are confusing and could be mistaken for the derivative of the function L(x), and o( ) in line 202 contradicts the standard infinitesimal notation.

There are instances where symbols are misused, such as \setminus in line 177, Eq. (6), fractional form in line 187, and o( ) in line 202, and unnecessary empty lines precede certain equations, just like Eq. 7, 9, 14, 16, 17.

The authors give two theorems for different settings, each with various norms, smoothness, and output designs. However, the relationship between these two theorems is not clear. It would be better to show that one is a corollary of the other.

Besides, I hope the authors could find more considerable advantages of this unique neuron through the theoretical investigation, which would strengthen the paper’s contributions.

**Limitations:**

The authors do discuss the limitations in Section 5. However, due to the issue with the parameter n (See Weaknesses), I think the discussion is insufficient. Besides, they mention that the theoretical results wouldn’t directly lead to an efficient approximation, so I wonder if there are any numerical experiments conducted based on the setting in this paper. Besides, how large is the gap between the theoretical setting and practical applications since this paper has many assumptions and hypotheses?

---

> ### Author Rebuttal · Authors · 2023-08-09
>
> We thank the reviewer for their time and their valuable feedback. While we are naturally disappointed by the outcome, we value the reviewer's expertise and the insights they provided. We answer here to the reviewer's comments, hoping they could change their idea about this work.
>
> > While the result is somewhat interesting, it fails [...] weakening its significance and attractiveness as well as this work.
>
> As summarized in the paper, the aim of this study is to establish if a MAM-based network can function as a universal approximator and we did so by providing theorems and a rigorous proof process. This is an initial step of paramount importance, to gain a theoretical understanding of MAM neurons, paving the way for further exploration. While we agree with the reviewer that a theoretical understanding of other properties of this neuron (e.g., prunability) is of interest, we believe that undertaking such an investigation would constitute a distinct and subsequent step in our research.
>
> > In Section 5, [...] a potential contradiction in this context.
>
> The reviewer's observation is valid, as the parameter $n$ is directly related to the scale of the network.
> The link between the approximation guarantee and the number of neurons is there and depends on two quantities that are uniformly indicated with $\delta$ and $\ell$. We will make this link explicit.
>
> $\delta$ (Theorem 1, see proof of Theorem 1) and $\delta$ and $\ell$ (Theorem 2, see Lemma 4) control the quality of approximation, but also the number of neurons needed in the first layer and implicitly in the second one (see the specialization of the general network construction in lines 141-143 for Theorem 1 and 161-173 for Theorem 2).
>
> The reviewer is correct in claiming that the theorems impose no constraints on the layer width. However, our intention was to emphasize a limitation rather than presenting it as an advantage. In fact, we do not impose any constraint on the magnitude of $n$ and allow it to be sufficiently large, which has repercussions on the efficiency of the approximation. We will rephrase "Limitations" to provide a clearer explanation.
>
> > There are several evident traces of incompleteness [...] just like Eq. 7, 9, 14, 16, 17.
>
> We apologize for any confusion caused by our stylistic choices.
> However, after careful proofreading, we may guarantee that none of the portions of text indicated by the reviewer contains any error or is grammatically incomplete, and only some minor typos are contained in a few equations:
> * *lines 141-145*, *equation after line 170*, *fractional form in line 187*: to the best of our knowledge, nothing wrong here.
> * *equation after line 216*: there is a minor typo and it will be fixed.
> * *Conclusions section*: as in other papers of this kind (e.g., [Lu, Yulong; Lu, Jianfeng. A Universal Approximation Theorem of Deep Neural Networks for Expressing Probability Distributions, NeurIPS (2020)]), conclusion is concise and can be extended but neither unfinished nor incorrect.
> * *lack of notation related to the width of the hidden layers*: we explained this above.
> * *norm defined after line 81*: indeed, that should be the Sobolev norm. However, there is a typo that may have misled the reviewer. Precisely, the term $\frac{\partial\phi}{x_j}(x)$ should have been written as $\frac{\partial\phi}{\partial x_j}(x)$.
> * *L'(x) and L''(x) in line 74*: in this work, first or second derivatives are always indicated as a fraction of differentials whereas $\mathcal{L}'$ and $\mathcal{L}''$ are defined in line 74 as the MAM hidden layers. We will revise the notation to avoid this small chance for ambiguity.
> * *o( ) in line 202*: the function $o(\cdot)$ is clearly defined right after its use. However, to avoid ambiguities, we will change the notation.
> * *$\setminus$ in line 177*: to the best of our knowledge, the LaTeX symbol \setminus $\setminus$ is commonly used to indicate the difference between two sets as in line 177.
> * *unnecessary empty lines*: we will remove the space between the text and the equations.
>
> > The authors give two theorems [...] one is a corollary of the other.
>
> The theorems are not one the corollary of the other. In our view, they are complementary and address in a different way the threefold trade-off between the complexity of the network, the strength of the error norm, and the smoothness requirements on the target function.
>
> Theorem 1 guarantees the strongest uniform approximation of targets, that are only required to be continuous. However, it requires a more complicated last layer that needs normalization.
>
> Theorem 2 guarantees a looser approximation as its error norm is of the integral type and thus allows local deviation in vanishing-measure neighborhood. Yet, this weaker approximation, paired with a stronger smoothness assumption on the target, allows us to leverage a simpler, purely linear, last layer and to show that not only the value of the target can be reproduced but also its first-order differential behavior.
>
> We agree with the reviewer on the opportunity of clarifying this in the paper and we will do that if accepted.
>
> > I wonder if there are any numerical experiments conducted [...]  this paper has many assumptions and hypotheses?
>
> We do not believe that our paper has many assumptions and hypotheses, as stated by the reviewer, since the only assumption is that the function to be approximated must be $\mathcal{C}^0$ for Theorem 1 and $\mathcal{C}^2$ for Theorem 2. Moreover, even if it is very uncommon in paper discussing the universal approximation property of neural networks, we did conduct numerical experiments to further verify the correctness of the theory and in the "Examples" section a numerical example is presented to enable the reader to visualize the difference between the two theorems. If the paper is accepted, we will provide more quantitative results about numerical experiments for different values of $n$ together with quantitative evaluations of the approximation error.

---

> > ### Comment · Reviewer_Ar7s · 2023-08-16
> >
> > I would like to thank the authors for their response. However, as noted by Reviewer p1mV, there might be simple proofs of the theorems. In addition, I found there are some highly related existing works, such as [1,2,3] below, which are absent in your paper. For example, [1] shows that GroupSort architectures are universal approximators. As there is a straightforward connection between max/min and sort, I tend to think of your results as a minor improvement of those results.  I agree with Reviewer p1mV that it is better for a lesser conference.
> >
> > [1] Anil, et al., Sorting Out Lipschitz Function Approximation, ICML, 2019
> > [2] Tanielian, et al., Approximating Lipschitz continuous functions with GroupSort neural networks, AISTATS 2021
> > [3] Bohang Zhang, et al., Rethinking Lipschitz Neural Networks for Certified L-infinity Robustness, NeurIPS 2022

---

> > > ### Author Response · Authors · 2023-08-20
> > >
> > > Though we cannot but thank the reviewer for the additional time spent on our paper, we would like to point out two issues that in our view are fundamental.
> > > Due to time constraints, they are laid down in a very synthetic and dry style.
> > >
> > > 1) we replied to the first round comment by highlighting that
> > >   1.a) they were mainly concentrated on (mostly nonexistent) formal issues
> > >   2.a) the technical problems were only apparent and derived from a very fast scan of the paper that, for example, led the reviewer to think that two distinct theorems (different assumptions, different guarantees) were the corollary one of the other or complaining that no discussion on the size of the network was present when all (though this is not the main topic of the paper as clearly declared from the beginning) elements are implicit in the proofs at the end of the paper and a very light revision would be able to explicit them
> > >
> > >
> > > To these clarification we received no reply and thus we are led to think that all these points are cleared. Yet
> > >
> > >
> > >
> > >
> > >
> > > 2) in this second round, this reviewer borrows from another reviewer that our result could be simply derived from other architectures already presented and discussed. Three papers are indicated. [1] Anil, et al., Sorting Out Lipschitz Function Approximation, ICML, 2019 [2] Tanielian, et al., Approximating Lipschitz continuous functions with GroupSort neural networks, AISTATS 2021 [3] Bohang Zhang, et al., Rethinking Lipschitz Neural Networks for Certified L-infinity Robustness, NeurIPS 2022
> > >
> > >
> > >
> > > Leaving alone the very serious procedural problem of changing first-round comments into completely different second-round comments borrowed from another reviewer, the technical point is completely void.
> > >
> > >
> > >
> > > In fact,  GroupSort *ACTIVATIONS* surely have universal approximation properties and in this sense they are equivalent to MAM and many other architecturs, yet, they achieve this with a scheme that is structurally different from MAM.
> > >
> > >
> > >
> > > In fact, even a fast scan of our paper and of those mentioned by the reviewer immediately highlights that GroupSort acts on the array of pre-activations, i.e. on a set of linear combinations of the layer inputs. Such linear combinations are key in building the architecture.
> > >
> > >
> > >
> > > In MAM we do not have any linear combination. The Max and Min operators are applied to separate weigthed layer inputs and Max and Min provide the aggregating part of the computation, while the activation is the classical ReLU. This MACless structure is what makes MAM so good in pruning and cannot be avoided.
> > >
> > >
> > >
> > > For this elementary reason, it is not at all straightforward to reproduce MAM behaviour with GroupSort. Actually the very high-level intuitive relationship between "sorting" and "max" and "min" is not a mathematical proof and though the suggested papers suggest that GroupSort can be specialize in something called MaxMin, the latter has nothing to do with MAM.
> > >
> > >
> > >
> > > In fact, the very same reviever provides not even a sketch of the path leading to the alleged equivalence.
> > >
> > >
> > >
> > > Add to the fact that MAM universality cannot be simply derived from GroupSort universality the fact that our Theorem 2 provides universal approximation of differential features of the target.
> > >
> > >
> > >
> > > In the light of all this we do believe that labeling our result as incremental is technically wrong and should not be accepted as the basis for a decision.

---

> > > > ### Comment · Reviewer_Ar7s · 2023-08-21
> > > >
> > > > I really don't want to take the time to care about the meaningless parts in the reply.
> > > >
> > > > 1. In the first paragraph of 'weakness', we clearly stated our main concerns: `the theorems and the proof process seem to have no investigation about these properties (allow more aggressive pruning)` . The authors' reply confirmed this. So we can conclude that the authors consider an interesting property of pruning, but the theorems are not about this. Therefore, I turn to hope to see if the theorem is new in the second reply.
> > > > 2. As noted, when we turn to the theorem itself, there are many related works absent. So the innovation of this paper can be reasonably questioned.
> > > > 3. When we go further, the MAM is *highly* related to sort, such as GroupSort.  So I tend to think of the results as a minor improvement and suggest a lesser conference.
> > > >
> > > > I think the only information in Official Comment by Authors is that: GroupSort used linear combinations but MAM does not have any linear combinations, so MAM and GroupSort are different. I agree they're not exactly the same, otherwise, your Th1 is a corollary rather than an improvement. However, I don't think the difference between with and without 'linear combination' is huge, because you used linear combination at the final layer and many combinations can be transferred back to the hidden layers.
> > > >
> > > > I hope the author can go deep into the literature of NN UAP rather than with a *fast scan*, and I will be happy to see the theoretic results on the pruning property of MAM.
> > > >
> > > > The last thing:  The decision is in AC's hands.

---

### Official Review · Reviewer_A9Cg · 2023-07-05

**Soundness:** 3 good
**Presentation:** 2 fair
**Contribution:** 3 good
**Rating:** 6
**Confidence:** 5

**Summary:**

This paper presents two universal approximation theorems for deep neural networks associated with a so-called Multiply-And-Max/min (MAM) activation function defined with the maximum and minimum of the input components and a bias constant. One is for uniform approximation and the other for approximation in the Sobolev space W^{1, p} with p\ge 1.

**Strengths:**

The presented universal approximation theorems are interesting and should have implications to show the power of pooling layers in deep neural networks. Studying simultaneous approximation in terms of the norm in the Sobolev space W^{1, p} should be able to explain the efficiency of some deep learning algorithms. These ideas are novel in my opinion.

**Weaknesses:**

 Though the approximation theorems are novel to me, the paper has a few weak points and should be improved:

1. To demonstrate some theoretical advantages of the MAM activation function. This might be done with the pooling layers in deep neural networks.

2. To present rates of approximation. Quantitative estimates for approximation of functions in various function spaces are crucial in the approximation error estimate for generalization analysis of deep learning algorithms.

3. To give rigorous statements and proofs. In Lemma 1, the sentence "Let z be any ... layer." should be removed because the output z is constructed by (5) and is not an arbitrary output function. In its proof, "We assume ..." and "the output ... only one of the inputs" should be revised: the neurons are constructed by (7), not by assumption. In Lemma 4, P is not a constant. It is a quantity depending on \ell of the form constant + o(1/\ell).

4. To give fair credits to the existing literature. For example, the construction of trapezoid functions has a long history in the study of deep neural networks and can be found in the papers of Shaham-Cloninger-Coifman (2018), Chui-Lin-Zhang-Zhou (2020), and some others.



**Questions:**

Some questions listed in the previous section should be answered.

**Limitations:**

Better theoretical results would improve the quality of the paper.

---

> ### Author Rebuttal · Authors · 2023-08-09
>
> We sincerely appreciate the time and valuable feedback provided by the reviewer. Below, we respond to the reviewer's comments, incorporating their insights to improve the quality of our work.
>
> > To demonstrate some theoretical advantages of the MAM activation function. This might be done with the pooling layers in deep neural networks.
>
> This is an interesting point, but as stated in the Introduction, this work is a first step in the definition of the theory behind MAM neurons. We decided to start with, from our point of view, the most important property that a neural network must guarantee, i.e. the universal approximation property. A theoretical demonstration about the advantages of MAM neurons presented in reference 1 of the paper will be the focus of a future work. Just as a note, Multiply-And-Max/min is a paradigm for a neuron and it is not an activation function (i.e., we use the ReLU activation function in our network). The idea of doing a comparison with pooling layers can be exploited. We may also highlight that even if the behavior of a MAM fully-connected layer may remind the one of a max-pooling layer, a MAM layer can be trained and the dimension of its output depends only on the number of neurons of the layer (i.e., the output could also contain more elements than the input).
>
> > To present rates of approximation. Quantitative estimates for approximation of functions in various function spaces are crucial in the approximation error estimate for generalization analysis of deep learning algorithms.
>
> We agree with the reviewer on the opportunity of adding info on the error estimate. We will do that in the final version of the paper, if accepted. Actually, error estimations are implicit in the proofs and depend on some key quantities: some bounding constant deriving from the smoothness assumptions we make on the target function, and an infinitesimal factor depending on the construction and thus, ultimately, on the number of neurons in the network layers.
>
> The estimate for Theorem 1 is implicit in the equation after line 158 and is substantially the modulus of continuity of the target function.
>
> The estimate for Theorem 2 is slightly more complicated and it has two parts.
> Due to the piecewise-linear nature of min and max functions, the scheme we adopt is clearly related to piecewise-linear interpolation that is used in disjoint hyperrectangular domains that can be made to fill the whole domain as well as in the regions between such domains, whose measure can be made to vanish not to contribute to the integral norm.
>
> Both terms depend on the features of the target function and on the construction of the network that administers the positioning and size of the hyperrectangular domains.
>
> In both cases we may apply classical error bound for piecewise-linear approximation that yields an estimate deriving from the statement of Lemma 4, in which one may relate the parameters $\delta$ and $\ell$ to the number of neurons in the network.
>
> > To give rigorous statements and proofs. In Lemma 1, the sentence "Let z be any ... layer." should be removed because the output z is constructed by (5) and is not an arbitrary output function. In its proof, "We assume ..." and "the output ... only one of the inputs" should be revised: the neurons are constructed by (7), not by assumption. In Lemma 4, P is not a constant. It is a quantity depending on $\ell$ of the form constant + o(1/$\ell$).
>
> We will coherently modify the sentence the reviewer has highlighted from Lemma 1.
> We also acknowledge that the proof of Lemma 4 is concise, leading the reader to mistakenly consider $P$ dependent on $\ell$. However, $P$ is actually a constant independent of $\ell$. To demonstrate it, we include below a comprehensive breakdown of all the intermediate steps starting from the equation at line 218 and leading to the conclusion of Lemma 4 and the definition of $P$.
>
> $$
> \begin{aligned}
> E_{\omega}
> &\le \left\[ \left(\frac{1}{2}M_2N^2\ell^2\right)^p + \left(\frac{1}{2} M_2N\ell\right)^p \right](2\ell)^N + \left[M_3^p+(2M_1)^p\right\] \left\[(2\ell+2\delta)^N-(2\ell)^N\right\]\\\\
> &= \left\[\left(\frac{1}{2}M_2N^2\ell\right)^p+\left(\frac{1}{2}M_2N\right)^p \right]\ell^p(2\ell)^{N}+\left[M_3^p+(2M_1)^p\right\] \left\[(2\ell+2\delta)^N-(2\ell)^N\right\]
> \end{aligned}
> $$
>
> Since $\ell\le 1$, the term $\left(\frac{1}{2}M_2N^2\ell\right)^p+ \left(\frac{1}{2}M_2N\right)^p$ is bounded above by $P=\left(\frac{1}{2}M_2N^2\right)^p+ \left(\frac{1}{2}M_2N\right)^p$. Here, we can also define $Q = \left[M_3^p+(2M_1)^p\right]$ so that we obtain:
>
> $$
> \begin{aligned}
> E_{\omega}
> &\le P \ell^p(2\ell)^{N} + Q \left\[(2\ell+2\delta)^N-(2\ell)^N\right\]\\\\
> &= (2\ell+2\delta)^N \left\\{P \ell^p \frac{(2\ell)^{N}}{(2\ell+2\delta)^N}+ Q \left\[1-\frac{(2\ell)^N}{(2\ell + 2\delta)^N}\right\] \right\\}\\\\
> &= (2\ell+2\delta)^N \left\\\{P \ell^p \frac{1}{(1 + \delta/\ell)^N} + Q \left\[1-\frac{1}{(1 + \delta/\ell)^N}\right\]\right\\\}\\\\
> &= (2\ell+2\delta)^N \left\\\{P \ell^p \left\[1 - o(\delta/\ell) \right\] + Q o(\delta/\ell)\right\\\}
> \end{aligned}
> $$
>
> with $o(\cdot) = 1 - 1/(1 + \cdot)^N$.
>
> Hence, $P$ is a constant that does not depend on $\ell$. We admit that the step where $P$ is defined as the upper bound of a term depending on $\ell$ was not intuitive and its lack may mislead the reader. For this reason, we will add it to the final version of the manuscript.
>
> > To give fair credits to the existing literature. For example, the construction of trapezoid functions has a long history in the study of deep neural networks and can be found in the papers of Shaham-Cloninger-Coifman (2018), Chui-Lin-Zhang-Zhou (2020), and some others.
>
> We are sorry for not mentioning these works and we thank you for the suggestion. We will be happy to add these references to the paper.

---

### Official Review · Reviewer_rhbL · 2023-07-06

**Soundness:** 4 excellent
**Presentation:** 3 good
**Contribution:** 3 good
**Rating:** 6
**Confidence:** 4

**Summary:**

The paper studies the universal approximation properties of ReLU networks using the Multiply-And-Max/min (MAM) neurons. Literature on the universal approximation properties of ReLU networks using the Multiply-and-ACcumulate (MAC) neurons is vast. However, the study on MAM neurons seems lacking. Hence, two theorems taking a step in characterizing the universal approximation properties for MAM neurons are proved in this paper. The first theorem states that a two-hidden-layer ReLU network using MAM neurons in the first two layers and the normalized linear combination in the last layer can approximate any continuous function on a unit hypercube arbitrarily well in terms of the infinity norm. The second theorem is similar to the first one, stating that a two-hidden-layer ReLU network using MAM neurons in the first two layers and the linear combination in the last layer can approximate any twice continuously differentiable function on a unit hypercube arbitrarily well in terms of the Sobolev norm. The proofs of these two theorems are constructive and the authors also acknowledge that their results do not imply efficient approximation.

**Strengths:**

The novelty is clear. The novel contribution of this paper is apparently the theoretical guarantees on the universal approximation properties of ReLU networks using MAM neurons. Although I have not carefully validated the proof, the explanations and statements given in the paper seem to be sufficiently convincing. Overall, this is a well-written paper. The presentation is concise, clear, and easy to follow. I enjoy reading the paper.

**Weaknesses:**

1. The requirement of the target function being twice continuously differentiable seems a bit limited. It would be great if the authors could relax this assumption or clarify why this assumption is necessary.
2. The result in Theorem 2 relies on the L^p Sobolev norm. Would it be possible to extend the result to the infinity norm? The paper would be more convincing and clearer if the authors can justify why the normalized linear combination and the linear combination use different norms. The connection between Theorem 1 and 2 seems missing.
3. Given the observation that using MAM or the mixed MAM/MAC neurons gives better pruning performance than the MAC neurons in practice, the paper would be more convincing if the authors can provide some insights into the constructive approximation of these different schemes.

**Questions:**

Line 81-82: Is the second term a partial derivative of phi with respect to x_j? There is a missing partial symbol in the denominator.

Line 112: The authors mention “...weakly unimodal piecewise-linear functions…” What does the word “weakly” mean here? Can we use unimodal piecewise-linear functions?

The two theorems provided in the paper rely on the assumption of using a two-hidden-layer network. I think it may be trivial to extend the results to multiple layers. Can we extend the results to a single-hidden-layer network? If not, would it be possible to prove it? It would be nice to give some insights into this.

**Limitations:**

The authors clearly state the limitations of their work in Section 5. Specifically, they point out that their results do not imply efficient implementation. I’m glad to see they make it very clear. They also state that the efficiency of approximation will be their future focus. I think this paper has laid a good foundation for their future work.

---

> ### Author Rebuttal · Authors · 2023-08-09
>
> We sincerely appreciate the time and valuable feedback provided by the reviewer. Below, we respond to the reviewer's comments, incorporating their insights to improve the quality of our work.
>
> > The requirement of the target [...] clarify why this assumption is necessary.
>
> We can clarify the motivation behind this assumption. The $\mathcal{C}^2$ assumption is used to leverage the Taylor expansion of the target function in equations (14) and (15). To prove the approximation of first-order derivative we need an expansion up to the same order with a remainder that can be bounded and pushed to zero by narrowing the neighborhood.
>
> This could be achieved locally by assuming that $f$ is only $\mathcal{C}^1$ if one gives up a well defined form for the remainder. This is surely possible, but i) it spoils the uniformity of the remainder over the whole domain and requires a more sophisticated bounding and ii) prevents the possibility of giving an easy error estimation depending on the size of the network.
>
> Since another reviewer asked for ii) we would keep the $\mathcal{C}^2$ assumption to be able to provide this insight.
>
> > The result in Theorem 2 relies [...] Theorem 1 and 2 seems missing.
>
> In our view, the theorems are complementary and address in a different way the threefold trade-off between the complexity of the network, the strength of the error norm, and the smoothness requirements on the target function.
>
> Theorem 1 guarantees the strongest uniform approximation of targets, that are only required to be continuous. However, it requires a more complicated last layer that needs normalization.
>
> Theorem 2 guarantees a looser approximation as its error norm is of the integral type and thus allows local deviation in vanishing-measure neighborhood. This weaker approximation, paired with a stronger smoothness assumption, allows us to leverage a simpler linear last layer and to show that both the value and its first-order differential behaviour can be reproduced.
>
> Despite potential smarter constructions, the transition from Sobolev to infinity norm is not a straightforward step.
> If we renounce to the normalized layer, we cannot ensure that the linear combination of the functions $z$ remains constant. To address this, we use a construction that deals with dimensionality effects and leaves certain areas of the domain uncovered. If the reviewer is interested, we would be glad to engage later in a further discussion on this topic.
>
> > Given the observation [...] different schemes.
>
> While this is an interesting observation, at present, we do not possess a theoretical foundation to clarify why neural networks based on MAM exhibit superior pruning performance. Nevertheless, there is empirical evidence of this, as documented in reference 1 and 2 of the paper. These studies show that, while maintaining a certain level of accuracy, using MAM neurons significantly increases the pruned connections compared to traditional MAC-only networks.
>
> The initial phase of our investigation involves examining whether a MAM-based network can function as a universal approximator. Thus, the primary objective of this work is to establish the universal approximation property for MAM networks. Addressing the pruning capability would necessitate a separate investigation and will be a subject of future work.
>
> > The authors mention “...weakly unimodal [...] unimodal piecewise-linear functions?
>
> The whole construction in the two theorems hinges on the availability of some basic building blocks that i) are functions of all the inputs, ii) are piecewise linear and can be obtained by MAM operations, iii) feature a maximum and are at least non-increasing in all directions departing from that maximum.
>
> The word *weakly* is used to signify the *non-increasing* profile away from the maximum and is needed as in many steps of our construction single-maximum functions would not fit.
>
> > The two theorems provided [...] some insights into this.
>
> We agree with the reviewer that extending our Theorems to more than two hidden layers would be trivial ad it depends only on the ability of encoding the identity in a MAM layer.
>
> Yet, though we do not have any formal proof yet, we believe that the same guarantee cannot be given for one-hidden-layer networks.
>
> Our assumption is supported from the classical model that is adopted for multi-input, one-layer networks that can be synthesized into an input-output relationship of the kind (see [Hornik “Approximation Capabilities of Multilayer Feedforward Networks”, Neural Networks, 1992])
>
> $$
> \sum_j y_j \Psi\left(\left\langle a_j,x\right\rangle+b_j \right)
> $$
>
> where $x$ is the input vector, $\Psi$ is the non-linearity, $y_j$ are the coefficients of the output linear layer, $a_j$ is the $j$-th coefficient vector, $b_j$ is the $j$-th bias and $\left\langle \cdot,\cdot\right\rangle$ stands for the scalar product.
>
> Regrettably, if one wants to use MAM neurons in all but the last layer, then the scalar product cannot be used.
>
> As the scalar product is out of the reach of a MAM neuron, the aggregation of the contributions coming from the processing of each input (equation (8) in the paper) must be done *after* some quantities depending separately on each input are computed in the first layer (equation (7) in the paper), and thus in a second hidden layer.
>
> Actually, a deeper analysis of equation (7) in the paper reveals that the MAM neurons in the first layer are used well below their potential computational capabilities: they basically implement a scaled ReLU with a variable bias.
> Hence, the construction uses only "*half*" of the first layer though we have to declare that two layers are involved.
>
> In the light of this we believe that reducing the number of hidden layers from 2 (one may even say  1.5 layers since the functionality of the neurons of the first layer are not fully exploited) to 1 is not a straightforward step though surely one worth analyzing in more detail to at least provide a formal negative result.

---

> > ### Comment · Reviewer_rhbL · 2023-08-21
> >
> > I would like to thank the authors for their responses. My concerns are addressed and I will keep my rating unchanged. However, the concern raised by Reviewer Ar7s is critical. I also agree that there is an apparent connection between sorting and max/min. Also, I found another related work that is missing in the manuscript [1]. Since any continuous function on a compact subset of $\mathbb{R}^n$ can be approximated by a continuous piecewise linear (CPWL) function and any CPWL function has a max-min representation, i.e.,
> > $p(x)=\max_{\mathcal{X}\in Q}\min_{i\in \mathcal{A}(\mathcal{X})}f_i(x)$, it is important to point out that the MAM neurons are a natural choice to approximating any CPWL function. In a very recent NeurIPS 2022 paper [1], it was proved that using a max-min representation can lead to better upper bounds on the number of neurons for representing CPWL functions.
> > From this, the universal approximation property of MAM neurons then seems to be trivial.
> >
> > [1] Chen, Kuan-Lin, Harinath Garudadri, and Bhaskar D. Rao. "Improved bounds on neural complexity for representing piecewise linear functions." Advances in Neural Information Processing Systems 35 (2022): 7167-7180.

---

> > > ### Author Response · Authors · 2023-08-21
> > >
> > > Thank you very much for appreciating our replies.
> > >
> > >
> > >
> > > Though the time window for interaction is closing, we would like to clarify that the concern of Ar7s is actually void.
> > >
> > >
> > >
> > > Regrettably, to realize why, it is necessary to overcome the, only apparent, analogy between some of the architectures already proposed that exploit max and min and our MAM neuron.
> > >
> > >
> > >
> > > As a matter of fact, the paper was probably not clear enough on this point and regrettably this led to a big misunderstanding.
> > >
> > >
> > >
> > > Starting from the paper that you suggest (the same happens in all other papers that have been mentioned as possible prior-art from which our result would *easily* follow) and considering, for example, eq 13, it is clear that, before min and max are applied, some functions $f_i$ have to be computed.
> > >
> > >
> > >
> > > Such functions are affine functions from R^N (N being the number of inputs) to R and thus are some kind of biased linear combination of the inputs.
> > >
> > >
> > >
> > > This does not happen in MAM structures that do not have any linear combination before min and max.
> > >
> > >
> > >
> > > More in neural terms and intuitively speaking, multi-input neurons are made of two stages. The first one weights and aggregates the inputs, the second decides the activation and thus the output of the neuron based on the aggregated pre-activation (possibly on the aggregated pre-activations of the whole layer as in the GroupSort case). All the examples and the literature that have been pointed out, play with the activation part (this is explicit when speaking of GroupSort but structurally the same for eq. 13 in [1]) and not with the aggregation part that, instead, is the one on which MAM focuses.
> > >
> > >
> > >
> > > This happens as MAM has been designed for pruning, a task for which is fundamental to assess the importance of each input before it is aggregated, an importance that is partially lost when applying a linear combination and considering only its result from there on.
> > >
> > >
> > >
> > > Regrettably, the unavailability of a linear combinatiom but in the last layer, prevents MAM networks from the possibility of tackling the problem of multi-dimensional input as the multi-dimensional profiles must be built by a suitable composition of nonlinear beahviours. This is actually what our second layer does.
> > >
> > >
> > >
> > > Clearly, if one addresses only one-input networks, such a complexity does not appear and this is what misled reviewer p1mV into believing that our result can be derived trivially from existing theory. It is not true for more than one input, i.e., in all the interesting cases.
> > >
> > >
> > >
> > > Overall, though we admit that explicitly mentioning in the paper some loosely related literature with suggestively similar title would have avoided all this, we are very sorry to see that this miscomprehension is heavily and negatively biasing the whole review process.

---

> > > > ### Comment · Reviewer_rhbL · 2023-08-21
> > > >
> > > > Thank you for your clarifications. Indeed, the architecture considered in the manuscript does not require a linear combination at the first layer. This is the reason why a linear combination (the parameters $c_k$) is required at the last layer. From this, I tend to agree with the authors that the previous concern of Ar7s is not that significant. Although the linear combination at the last layer can be transferred back to the first layer, this process seems nontrivial and new. In addition, this may be the key why MAM neurons allow aggressive pruning, as pointed out by the authors in the reply. I think this is a very important difference compared to existing works like [1] and may be used to partially address the concern raised by Reviewer Ar7s regarding some results of pruning MAM neurons. I would encourage the authors to include the discussion of this matter in their manuscript. I think the authors have done a good job in clarifying the questions from the reviewers, and I appreciate the effort and time that the authors have put in. Considering the discussion, I am inclined towards accepting the paper if the authors can enhance their manuscript as suggested, given that the results provided lay down the first step in understanding MAM neurons and their difference compared to MAC-based neurons.

---

### Decision · Program_Chairs · 2023-09-21

**Decision:**

Reject

**Comment:**

Authors established new theory on a promising extrema nonlinear network that seems to be particularly useful for sparsification. Although novel and solid, the theory only provides hints towards explaining the pruning performance. The contributions are not of broad impact for NeurIPS at this moment, and perhaps better for another venue.